# Simplified Estimation Method of Plastic Energy Dissipation for MDOF Systems Using Force Analogy Method

**Yingna Mu** [1], **Jiting Qu** [2,*], **Yu Shu** [3] **and Yanbin Tan** [2]

1    Ocean and Civil Engineering Institute, Dalian Ocean University, Dalian 116023, China; myn-myn@dlou.edu.cn
2    School of Civil Engineering, Faculty of Infrastructure Engineering, Dalian University of Technology, Dalian 116024, China
3    China Northeast Architecture Design and Research Institute Co., Ltd., Dalian Branch, Dalian 116000, China
*    Correspondence: qjt@dlut.edu.cn; Tel.: +86-0411-84763479 or +86-186-0428-1847

**Abstract:** Plastic energy dissipation is a key factor in the response of inelastic structures subjected to seismic input, and is often regarded as a primary source of structural damage due to the inelastic deformation of structural components. Accurately predicting a structure's plastic energy dissipation is essential for efficient energy-based design and seismic assessment. While a single-degree-of-freedom (SDOF) system provides a simple and effective method for estimating plastic energy dissipation, few studies have explored the use of nonlinear analysis methods or equivalent SDOF systems for this purpose. Based on the principle-of-force-analogy method, firstly, the formulas of plastic energy dissipation of a multi-degree-of-freedom (MDOF) system and its equivalent SDOF system were established. Secondly, two estimation methods of plastic energy dissipation of MDOF systems were proposed. Finally, numerical simulations were performed on several multi-story and high-rise structures with varying heights and spans to compare the plastic energy dissipation of MDOF systems and the equivalent SDOF systems of different modes. The simulation results demonstrate that the proposed methods and formulas accurately estimate plastic energy dissipation in multi-story and high-rise structures, while also requiring fewer calculations and less storage.

**Keywords:** plastic energy dissipation; multi-degree-of-freedom system; equivalent single-degree-of-freedom system; force analogy method

## 1. Introduction

In 1956, Housner proposed an energy-based design method [1] that viewed the response of structures to earthquakes as a process of energy transformation and dissipation. Originally developed for the limit design of single-degree-of-freedom (SDOF) systems [1], this method has since gained extensive attention and become an applicable choice for the design of new structures and the seismic capacity assessment of existing buildings. Many researchers have studied the elastic–plastic displacement response of structures by analyzing energy spectra, and various energy spectra have been proposed [2–4].

When a structure is in an inelastic state, its input energy is mainly offset by plastic energy dissipation and damping energy dissipation [5]. Many researchers [6–8] have constructed a plastic energy dissipation spectrum to study the elastoplastic responses of a structure. However, most of the research works that use energy concepts to analyze structural seismic responses mainly focus on SDOF systems, and only a few have been extended to MDOF systems [9–11]. Real structures are usually simplified as MDOF systems. When performing energy analysis, due to the complicated process and a large amount of calculation, estimating the input and hysteretic energy of MDOF systems based on SDOF systems is often considered a simple and effective method [12–19]. The classical method of approximating plastic energy in structures is based on calculating the area enclosed by the force versus displacement response [20]. This accumulated plastic energy, caused

by nonlinear hysteresis cycles, is one of the most important factors that induce structural damage during earthquakes. However, obtaining the force versus displacement response in a MDOF system is challenging, as the force corresponding to each displacement degree of freedom is unknown. Due to limited research in this area, further investigations are required to determine the energy dissipation relationship between MDOF and equivalent SDOF systems.

The force analogy method (FAM) was first proposed by Lin [21] for the inelastic analysis in continuum mechanics. Wong and Yang [22,23] later extended this method to include the dynamic analysis of nonlinear steel frame structures, as well as predictive instantaneous optimal control and energy evaluation of inelastic systems. This method is particularly simple for calculating plastic energy. Over time, many scholars have not only worked to improve the FAM theory [24], but also explored the application of this method [25–27]. Hao et al. [28] presented the static pushover analysis for nonlinear fiber beam element conducted on the foundation of the FAM, and the results show that the algorithm complexity of the proposed method decreased about 80%, and its computing efficiency increased at least five times. The accumulation of research in this area indicates that the FAM is a reliable, accurate, and efficient method for analysis, since it only requires the initial stiffness of structures.

The purpose of this study is to establish formulas of plastic energy of MDOF systems and their equivalent SDOF systems using the FAM based on the assumption of the equivalent SDOF system and the mode decomposition method. The plastic energy of the MDOF systems and their equivalent SDOF systems were compared using several structures to verify the effectiveness and application conditions of the two types of formulas. This research aims to propose two estimation methods for the plastic energy dissipation of a MDOF system, which can serve as a useful tool to analyze structural damage for energy-based seismic design. In seismic design, the inelastic energy spectrum is often used, and by using the proposed method, the plastic energy spectrum can be obtained and be suitable for multi-degree-of-freedom systems, which will be introduced in another research article.

## 2. Formula of Plastic Energy Base on Force Analogy Method

The FAM is a method that utilizes inelastic deformation as the variable instead of the stiffness. This method allows for the initial stiffness matrix to be used throughout the inelastic analysis, resulting in high efficiency and accuracy in calculation. Wong and Yang [14] presented the principle of the FAM in detail during the analysis of braced frames; therefore, only a brief summary is provided as background information. Studying from a SDOF system point of view, the FAM analyzes the total displacement *x(t)* caused by external force $F_S(t)$ by dividing it into two parts: the elastic displacement *x'(t)* and the inelastic displacement *x''(t)*. In a bilinear force versus displacement relationship, the elastic line OA is elongated to point C, which corresponds to the force $F_S(t)$. The displacement of point C is defined as elastic displacement as shown in Figure 1. The relationship between the restoring force and displacement in the system is written as:

$$F_s = k_0 x'(t) = k_0(x(t) - x''(t)), \tag{1}$$

where $k_0$ represents the initial stiffness of the system.

The equation of motion for an n-degree-freedom structure subjected to an earthquake can be expressed as:

$$M_S \ddot{X}(t) + C\dot{X}(t) + K_e X'(t) = -M_S \ddot{X}_g(t) \tag{2}$$

where $M_S$, C, and $K_e$ denote the $n \times n$ mass, inherent damping, and initial stiffness matrices of the structure, respectively; $\dot{X}(t)$ and $\ddot{X}(t)$ denote the relative velocity and acceleration vector of the n-dimension; $X'(t)$ is the n-dimensional elastic displacement vector; and $\ddot{X}_g(t)$ is the seismic excitation.

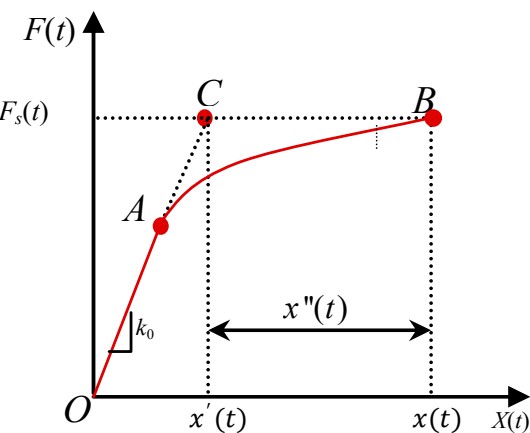

**Figure 1.** Bilinear relationship of force versus displacement.

Assuming that inelastic deformations of a structure are concentrated on the ends of columns and beams, these locations are defined as plastic hinge locations (PHL) and assigned as the plastic rotational degree of freedom (RDOF). Equation (2) can be simplified through static condensation by ignoring both rotational masses of inertia and rational damping at the RDOFs and can be expressed in the following form:

$$M_{dd}\ddot{X}_d(t) + C_{dd}\dot{X}_d(t) + \overline{K}X_d(t) = -M_{dd}I_l a_g(t) + \overline{K}X''_d(t) \tag{3}$$

where $I_l$ is a column vector, and its elements are all 1; $a_g$ is the acceleration of the ground. $\overline{K}$ represents the elastic stiffness matrix of a structure after performing static condensation,

$$\overline{K} = K_{dd} - K_{dr}K_{rr}^{-1}K_{rd}, \tag{4}$$

where the subscripts *d* and *r* indicate the matrices or vectors corresponding to the displacement DOFs and the rotation DOFs, respectively; for an *n* DOF system, *n* = *d* + *r*.

The total moments, $M(t)$ restoring forces $F_R(t)$, and restoring moments $M_R(t)$ caused by plastic rotation at these locations are written as:

$$M(t) = M'(t) + M''(t), \tag{5}$$

$$F_R(t) = -K_P\Theta''(t) = -K_e X''(t), \tag{6}$$

$$M_R(t) = -K_R\Theta''(t), \tag{7}$$

where $M(t)$, $M'(t)$, and $M''(t)$ denote the total moment, the elastic moment caused by elastic displacement, and the residual moment due to inelastic displacement; $K_P$ is the matrix related to the plastic rotation at PRDOFs with the applied force on the DOF of the system; $K_R$ represents the matrix related the plastic rotation with the moments at PRDOFs; and $\Theta''(t)$ is the plastic rotation vector.

The plastic moment vector due to inelastic placement becomes:

$$M''(t) = -\left((K_R - K_P^T K_e^{-1} K_P)\Theta''(t)\right), \tag{8}$$

The elastic moment vector related to elastic placement is:

$$M'(t) = K_P^T(X(t) - K_e^{-1}K_P\Theta''(t)), \tag{9}$$

Substituting Equations (8) and (9) into Equation (5), the total moment vector is:

$$M(t) = K_P^T X(t) - K_R \Theta''(t),\tag{10}$$

According to Equation (1), the force of a system can be written as:

$$F(t) = K_e X'(t) = K_e X(t) - K_P \Theta''(t),\tag{11}$$

The governing inelastic equation of a system based on the FAM is formulated according to Equations (10) and (11).

$$\left\{ \begin{array}{c} F(t) \\ M(t) \end{array} \right\} = \begin{bmatrix} K_e & K_P \\ K_P^T & K_R \end{bmatrix} \left\{ \begin{array}{c} X(t) \\ -\Theta''(t) \end{array} \right\},\tag{12}$$

The energy equation can be derived based on the FAM (discussed in detail in ref. [23]) as:

$$\int_0^{t_k} \ddot{Y}(t)^T M_S dY(t) + \int_0^{t_k} \dot{X}(t)^T C dX(t) + \int_0^{t_k} X'(t)^T K_e dX'(t) + \int_0^{t_k} X'(t)^T K_e dX''(t) = \int_0^{t_k} \ddot{Y}(t)^T M_S dX_g(t),\tag{13}$$

where $\ddot{Y}(t)$ is the absolute acceleration of a structure.

The plastic energy of a structure, denoted as $PE$, can be seen as the sum of energy dissipation due to plastic hinge rotations, as follows:

$$PE = \int_0^{t_k} X_d'^T \overline{K} dX_d'' = \int_0^{t_k} M'(t)^T d\Theta''(t) = \sum_{i=1}^{m} \int_0^{t_k} M_i'(t)^T d\Theta_i''(t) = \sum_{i=1}^{m} PE_i,\tag{14}$$

where $PE_i$ is the plastic energy at the $i$th PHL.

## 3. Two Plastic Energy Estimation Methods Based on an Equivalent SDOF System

There are not many research studies on the application of mode decomposition methods in energy response equations [29,30]. Although classical modal analysis is not applicable to inelastic systems, this idea has been used in modal pushover analysis for inelastic buildings [31]. An approximate response history analysis procedure was used as the basis for developing an estimation method for an inelastic system.

### 3.1. Estimation Method Considering Only the First Mode Shape

Based on the assumption of equivalent SDOF (ESDOF) systems [8], if the seismic response of a structure is mainly dominated by the ith mode shape and the mode shape remains constant throughout the response, the story displacement $X_d(t)$ of a structure can be approximated by multiplying the generalized displacement $q_i(t)$ with the $i$th shape function $\Phi_i$, as follows:

$$X_d(t) = q_i(t) \Phi_i,\tag{15}$$

where $\Phi_i$ represents the normalized mode shape, the modal value of the top floor equals 1, and $q_i(t)$ is the displacement of the top floor.

Based on the FAM concept, the generalized displacement can be divided into elastic and inelastic displacement as:

$$q_i(t) = q_i'(t) + q_i''(t),\tag{16}$$

$$X_d'(t) = q_i'(t) \Phi_i,\tag{17}$$

$$X_d''(t) = q_i''(t) \Phi_i,\tag{18}$$

where $q_i'(t)$ and $q_i''(t)$ represent an elastic and an inelastic generalized displacement, respectively.

Substituting Equations (15)–(18) into Equation (3) and multiplying on the left by $\boldsymbol{\Phi}_i^T$ provides:

$$\boldsymbol{\Phi}_i^T \boldsymbol{M_{dd}}\boldsymbol{\Phi}_i\ddot{q}(t) + \boldsymbol{\Phi}_i^T \boldsymbol{C_{dd}}\boldsymbol{\Phi}_i\dot{q}(t) + \boldsymbol{\Phi}_i^T \overline{\boldsymbol{K}}\boldsymbol{\Phi}_iq'(t) = -\boldsymbol{\Phi}_i^T \boldsymbol{M_{dd}}\boldsymbol{I_l}a_g(t) \tag{19}$$

The displacement of the ith order mode shape of equivalent SDOF system subjected to earthquake can be written as:

$$\xi(t) = q(t)/\Gamma_i, \tag{20}$$

where $\Gamma_i$ is the participation coefficient of the ith order mode shape,

$$\Gamma_i = \boldsymbol{\Phi}_i^T \boldsymbol{M_{dd}}\boldsymbol{I_l} / \boldsymbol{\Phi}_i^T \boldsymbol{M_{dd}}\boldsymbol{\Phi}_i \tag{21}$$

$\xi(t)$ can also be divided into two parts based on the FAM,

$$\xi(t) = \xi'(t) + \xi''(t), \tag{22}$$

$$\xi'(\mathrm{t}) = q'(t)/\Gamma_\mathrm{i}, \tag{23}$$

$$\xi''(t) = q''(t)/\Gamma_\mathrm{i}, \tag{24}$$

Substituting Equations (20)–(24) into Equation (19) provides:

$$\boldsymbol{\Phi}_i^T \boldsymbol{M_{dd}}\boldsymbol{I_l}\ddot{\xi}(t) + \boldsymbol{\Phi}_i^T \boldsymbol{C_{dd}}\boldsymbol{\Phi}_i\Gamma_i\dot{\xi}(t) + \boldsymbol{\Phi}_i^T \overline{\boldsymbol{K}}\boldsymbol{\Phi}_i\Gamma_i\xi'(t) = -\boldsymbol{\Phi}_i^T \boldsymbol{M_{dd}}\boldsymbol{I_l}a_g(t), \tag{25}$$

Equation (25) can be simplified as:

$$M_i^*(\ddot{\xi}(t) + a_g(t)) + C_i^*\dot{\xi}(t) + K_i^*\xi'(t) = 0, \tag{26}$$

where $M_i^* = \boldsymbol{\Phi}_i^T M_{dd}I_l$, $C_i^* = \boldsymbol{\Phi}_i^T C_{dd}\boldsymbol{\Phi}_i\Gamma_i$, and $K_i^* = \boldsymbol{\Phi}_i^T \overline{K}\boldsymbol{\Phi}_i\Gamma_i$ represent the equivalent mass, equivalent damping, and equivalent elastic stiffness of the *i*th order mode shape of equivalent SDOF system, respectively.

Therefore, the plastic energy of the *i*th order mode shape of the equivalent SDOF system can be expressed as:

$$PE^* = \int_0^{t_k} \xi'^T K_i^* d\xi'', \tag{27}$$

Substituting Equations (15)–(18) into Equation (14) provides:

$$PE = \int_0^{t_k} (\Phi_i q')^T \overline{K} d(\Phi_i q'') = \int_0^{t_k} q'^T \Phi_i^T \overline{K}\Phi_i dq'' \tag{28}$$

Combining Equation (25) with Equations (23) and (24), the *PE* can be written as:

$$PE = \int_0^{t_k} \Gamma_i^T \xi'^T \Phi_i^T \overline{K}\Phi_i d(\Gamma_i \xi'') = \int_0^{t_k} \Gamma_i^T \xi'^T \Phi_i^T \overline{K}\Phi_i \Gamma_i d\xi'' = \Gamma_i \int_0^{t_k} \xi'^T K_i^* d\xi'' = \Gamma_i PE^*, \tag{29}$$

When the seismic response of a structure is mainly based on the *i*th mode shape, the relationship of plastic energy dissipation between MDOF systems and equivalent SDOF systems can be expressed by Equation (29).

### 3.2. Estimation Method Based on the Mode Decomposition Method

According to the mode decomposition method, the story displacement of a MDOF structure can be calculated by multiplying the generalized displacement with the structure mode, as follows:

$$\boldsymbol{X_d}(t) = \boldsymbol{\Phi}\{Q(t)\} \tag{30}$$

where $\{Q(t)\}$ is the column vector of the generalized displacement, with each element in the vector being determined by the mode and displacement of the structure. $\Phi$ is the matrix of modal shape, with the $i$th column vector representing the $i$th mode shape.

Based on the FAM concept, the generalized displacement vector can be divided into elastic and inelastic displacement as:

$$\{Q(t)\} = \{Q'(t)\} + \{Q''(t)\}, \tag{31}$$

$$X'_d(t) = \Phi\{Q'(t)\}, \tag{32}$$

$$X''_d(t) = \boldsymbol{\Phi}\{Q''(t)\}, \tag{33}$$

Substituting Equations (30)–(33) into Equation (3) and multiplying on the left by the transpose matrix of modal shape $\Phi$ provides:

$$\boldsymbol{\Phi}^T\boldsymbol{M_{dd}}\boldsymbol{\Phi}\left\{\ddot{Q}(t)\right\} + \boldsymbol{\Phi}^T\boldsymbol{C_{dd}}\boldsymbol{\Phi}\left\{\dot{Q}(t)\right\} + \boldsymbol{\Phi}^T\overline{\boldsymbol{K}}\boldsymbol{\Phi}\{Q'(t)\} = -\boldsymbol{\Phi}^T\boldsymbol{M_{dd}}\boldsymbol{I_l}a_g(t), \tag{34}$$

The displacement of mode shape can be written as:

$$\{Q(t)\} = [R]\{\delta(t)\} = \begin{bmatrix} \Gamma_1 & 0 & \cdots & 0 & 0 \\ 0 & \Gamma_2 & \ddots & \ddots & 0 \\ \vdots & \ddots & \ddots & \ddots & \vdots \\ 0 & \ddots & \ddots & \ddots & 0 \\ 0 & 0 & \cdots & 0 & \Gamma_n \end{bmatrix} \begin{Bmatrix} \delta_1(t) \\ \delta_2(t) \\ \vdots \\ \vdots \\ \delta_n(t) \end{Bmatrix} \tag{35}$$

where $[R]$ is the matrix of mode participation coefficient and $\{\delta(t)\}$ is the displacement vector of mode shape of the equivalent SDOF system, which can also be divided into two parts based on the FAM.

$$\{\delta(t)\} = \{\delta'(t)\} + \{\delta''(t)\}, \tag{36}$$

$$\{\delta'(t)\} = [R]^{-1}\{Q'(t)\}, \tag{37}$$

$$\{\delta''(t)\} = [R]^{-1}\{Q''(t)\}, \tag{38}$$

Substituting Equations (35)–(38) into Equation (34) produces:

$$\boldsymbol{\Phi}^T\boldsymbol{M_{dd}}\boldsymbol{\Phi}[R]\left\{\ddot{\delta}(t)\right\} + \boldsymbol{\Phi}^T\boldsymbol{C_{dd}}\boldsymbol{\Phi}[R]\left\{\dot{\delta}(t)\right\} + \boldsymbol{\Phi}^T\overline{\boldsymbol{K}}\boldsymbol{\Phi}[R]\{\delta'(t)\} = -\boldsymbol{\Phi}^T\boldsymbol{M_{dd}}\boldsymbol{I_l}a_g(t), \tag{39}$$

where

$$\begin{aligned} \boldsymbol{\Phi}^T\mathbf{M_{dd}}\boldsymbol{\Phi}[R] &= \begin{bmatrix} \Phi_1 & \Phi_2 & \cdots & \cdots & \Phi_n \end{bmatrix}^T M_{dd} \begin{bmatrix} \Phi_1 & \Phi_2 & \cdots & \cdots & \Phi_n \end{bmatrix}[R] \\ &= \begin{bmatrix} \Phi_1^T M_{dd}\Phi_1 & \Phi_1^T M_{dd}\Phi_2 & \cdots & \Phi_1^T M_{dd}\Phi_{n-1} & \Phi_1^T M_{dd}\Phi_n \\ \Phi_2^T M_{dd}\Phi_1 & \Phi_2^T M_{dd}\Phi_2 & \ddots & \ddots & \Phi_2^T M_{dd}\Phi_n \\ \vdots & \ddots & \ddots & \ddots & \vdots \\ \Phi_{n-1}^T M_{dd}\Phi_1 & \ddots & \ddots & \ddots & \Phi_{n-1}^T M_{dd}\Phi_n \\ \Phi_n^T M_{dd}\Phi_1 & \Phi_n^T M_{dd}\Phi_2 & \cdots & \Phi_n^T M_{dd}\Phi_{n-1} & \Phi_n^T M_{dd}\Phi_n \end{bmatrix}[R] \end{aligned} \tag{40}$$

According to the first orthogonality of mode shapes, Equation (40) can be written in the form of:

$$\boldsymbol{\Phi}^T \boldsymbol{M_{dd}} \boldsymbol{\Phi}[R] = \begin{bmatrix} \Phi_1^T M_{dd} \Phi_1 \Gamma_1 & 0 & \cdots & 0 & 0 \\ 0 & \Phi_2^T M_{dd} \Phi_2 \Gamma_2 & \ddots & \ddots & 0 \\ \vdots & & \ddots & \ddots & \vdots \\ 0 & & \ddots & \ddots & 0 \\ 0 & 0 & \cdots & 0 & \Phi_n^T M_{dd} \Phi_n \Gamma_n \end{bmatrix}$$

$$= \begin{bmatrix} \Phi_1^T M_{dd} I_l & 0 & \cdots & 0 & 0 \\ 0 & \Phi_2^T M_{dd} I_l & \ddots & \ddots & 0 \\ \vdots & & \ddots & \ddots & \vdots \\ 0 & & \ddots & \ddots & 0 \\ 0 & 0 & \cdots & 0 & \Phi_n^T M_{dd} I_l \end{bmatrix} = \boldsymbol{M^*} \qquad (41)$$

where $\boldsymbol{M^*}$ is the equivalent mass matrix.

According to the second orthogonality of mode shapes, the third term on the left of Equation (39) can be simplified as:

$$\boldsymbol{\Phi}^T \overline{\boldsymbol{K}} \boldsymbol{\Phi}[R] = \begin{bmatrix} \Phi_1^T \overline{K} \Phi_1 \Gamma_1 & 0 & \cdots & 0 & 0 \\ 0 & \Phi_2^T \overline{K} \Phi_2 \Gamma_2 & \ddots & \ddots & 0 \\ \vdots & & \ddots & \ddots & \vdots \\ 0 & & \ddots & \ddots & 0 \\ 0 & 0 & \cdots & 0 & \Phi_n^T \overline{K} \Phi_n \Gamma_n \end{bmatrix} = \boldsymbol{K^*}, \qquad (42)$$

where $\boldsymbol{K^*}$ is the equivalent elastic stiffness matrix. Assuming that the damping matrix satisfies the orthogonality condition of the vibration mode, the second term on the left side of Equation (39) can be simplified as:

$$\boldsymbol{\Phi}^T \boldsymbol{C_{dd}} \boldsymbol{\Phi}[R] = \begin{bmatrix} \Phi_1^T C_{dd} \Phi_1 \Gamma_1 & 0 & \cdots & 0 & 0 \\ 0 & \Phi_2^T C_{dd} \Phi_2 \Gamma_2 & \ddots & \ddots & 0 \\ \vdots & & \ddots & \ddots & \vdots \\ 0 & & \ddots & \ddots & 0 \\ 0 & 0 & \cdots & 0 & \Phi_n^T C_{dd} \Phi_n \Gamma_n \end{bmatrix} = \boldsymbol{C^*}, \qquad (43)$$

where $\boldsymbol{C^*}$ is the equivalent damping matrix.

Thus, Equation (3) can be simplified as:

$$\begin{Bmatrix} M_1^* \ddot{\delta}_1(t) \\ M_2^* \ddot{\delta}_2(t) \\ \vdots \\ \vdots \\ M_n^* \ddot{\delta}_n(t) \end{Bmatrix} + \begin{Bmatrix} C_1^* \dot{\delta}_1(t) \\ C_2^* \dot{\delta}_2(t) \\ \vdots \\ \vdots \\ C_n^* \dot{\delta}_n(t) \end{Bmatrix} + \begin{Bmatrix} K_1^* \delta'_1(t) \\ K_2^* \delta'_2(t) \\ \vdots \\ \vdots \\ K_n^* \delta'_n(t) \end{Bmatrix} = -\begin{Bmatrix} M_1^* \\ M_2^* \\ \vdots \\ \vdots \\ M_n^* \end{Bmatrix} a_g(t), \qquad (44)$$

The above equation is the equation of motion for equivalent SDOF systems.

Substituting Equations (30)–(33) into Equation (14) provides:

$$PE = \int_0^{t_k} (\Phi\{Q'(t)\})^T \overline{K} d(\Phi\{Q''(t)\}) = \int_0^{t_k} \{Q'(t)\}^T \Phi^T \overline{K} \Phi d\{Q''(t)\}, \qquad (45)$$

Substituting Equations (35)–(38) into Equation (45), the relationship of plastic energy dissipation between MDOF systems and equivalent SDOF systems can be expressed as:

$$
\begin{aligned}
PE &= \int_0^{t_k} \{\delta'(t)\}^T [R]^T \Phi^T \overline{K} \Phi d\{[R]\{\delta''(t)\}\} \\
&= \int_0^{t_k} \{\delta'(t)\}^T [R]^T \Phi^T \overline{K} \Phi [R] d\{\delta''(t)\} \\
&= \int_0^{t_k} \{\delta'(t)\}^T [R]^T K^* d\{\delta''(t)\} \\
&= \int_0^{t_k} \sum_{i=1}^n \Gamma_i \delta_i'(t) K_i^* \dot{\delta}_i''(t) dt \\
&= \sum_{i=1}^n \int_0^{t_k} \Gamma_i \delta_i'(t) K_i^* \dot{\delta}_i''(t) dt = \sum_{i=1}^n PE2_i,
\end{aligned}
\tag{46}
$$

where $PE2_i$ is the plastic energy dissipation of the $i$th order mode shape of an equivalent SDOF system, given by:

$$
PE2_i = \int_{t=0}^{t=t_k} \Gamma_i \delta_i'(t) K_i^* \dot{\delta}_i''(t) dt,
\tag{47}
$$

The comparison of Equations (27) and (47) indicates that the expressions of the plastic energy dissipation of ESDOF systems defined by the two methods are different. The plastic energy dissipation of a MDOF system during earthquakes is equal to the sum of plastic energy dissipation of each order mode shape of an ESDOF system based on the mode decomposition method.

## 4. Numerical Analysis

### 4.1. Structure Models

Three steel frame structures were selected to illustrate the application of the proposed method and to verify the efficiency of the established relationship mentioned in Section 3.

Model 1 is a two-story steel frame with unequal span, and the serial number of columns and beams is shown in Figure 2a. The mass was $3.6 \times 10^5$ kg and $2.4 \times 10^5$ kg for the first and second floors, respectively. The stress–strain relation of steel was considered to be bilinear. Stiffness-proportional damping was adopted and the damping ratio was 0.05.

Model 2 is a five-story steel frame with three equal spans, and the serial number of columns and beams is shown in Figure 2b. The mass of each floor was $4 \times 10^5$ kg. A bilinear model was used for the stress–strain relation. Mass-proportional damping was adopted. The structure was dominated by multiple modes with a damping ratio of 0.04.

Model 3 is an eleven-story steel frame with different spans and heights, and the serial number of columns and beams is shown in Figure 2c. The masses of the first to the fifth floor were $9.0 \times 10^4$ kg, the masses of the sixth to the eighth floor were $5.5 \times 10^4$ kg, and the masses of the ninth to the eleventh floor were $2.5 \times 10^4$ kg. Mass-proportional damping was adopted and the damping ratio was 0.04.

The section parameters of the three structures are shown in Table 1 and structural modal parameters are summarized in Table 2.

**Table 1.** Section parameters of structural components.

| Model Number | Serial Number | 1 | 2 | 3 | 4 |
|---|---|---|---|---|---|
| 1 | Moment of inertia (N·m²) | $3.20 \times 10^8$ | $4.54 \times 10^8$ | $4.87 \times 10^7$ | $1.17 \times 10^8$ |
| | Yielding moment (kN·m) | 2210 | 2998 | 601 | 1445 |
| 2 | Moment of inertia (N·m²) | $3.20 \times 10^8$ | $4.54 \times 10^8$ | $2.0 \times 10^8$ | $6.99 \times 10^8$ |
| | Yielding moment (kN·m) | 2210 | 2998 | 1445 | 601 |
| 3 | Moment of inertia (N·m²) | $2.40 \times 10^8$ | $4.98 \times 10^8$ | $4.98 \times 10^8$ | $2.40 \times 10^8$ |
| | Yielding moment (kN·m) | 1250 | 2160 | 1730 | 1000 |

**Table 2.** Structural modal parameters of three structures.

| Model Number | Modal Order | Periods (s) | Frequency ($s^{-1}$) | Mode of Vibration |
|---|---|---|---|---|
| 1 | 1 | 1.20 | 5.2366 | 0.4396, 1 |
| | 2 | 0.36 | 17.653 | 1, −0.6594 |
| 2 | 1 | 1.97 | 3.1882 | 0.128, 0.377, 0.635, 0.856, 1 |
| | 2 | 0.57 | 10.968 | −0.458, −0.978, −0.871, −0.023, 1 |
| | 3 | 0.28 | 22.432 | −0.889, −0.900, 0.571, 1, −0.766 |
| | 4 | 0.17 | 37.391 | 1, −0.117, −0.857, 0.982, −0.380 |
| | 5 | 0.112 | 54.041 | −0.960, 1, −0.756, 0.394, −0.110 |
| 3 | 1 | 1.02 | 6.139 | - |
| | 2 | 0.43 | 14.675 | - |
| | 3 | 0.25 | 25.040 | - |
| | 4 | 0.17 | 36.030 | - |
| | 5 | 0.11 | 53.712 | - |
| | 6 | 0.09 | 68.600 | - |

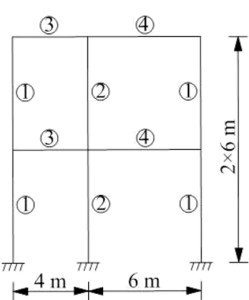
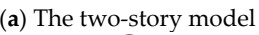

(**a**) The two-story model

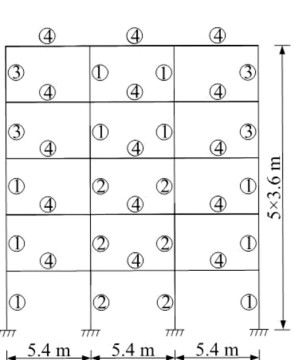

(**b**) The five-story model

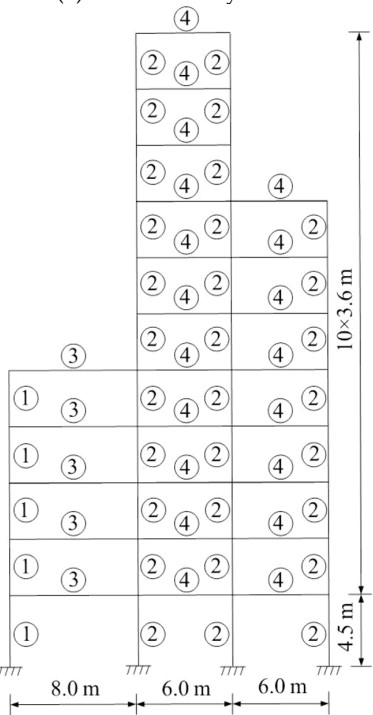

(**c**) The eleven-story model

**Figure 2.** Model 1–Model 3 (①–④ are component number).

### 4.2. Earthquake Records

Three earthquake records selected for the numerical analysis for the input earthquake accelerations to the structures were Morgan Hill, Kocaeli, and Kobe, as shown in Figure 3. The values of peak ground accelerations were all scaled to 400 cm/s², 500 cm/s², 600 cm/s², and 700 cm/s² respectively.

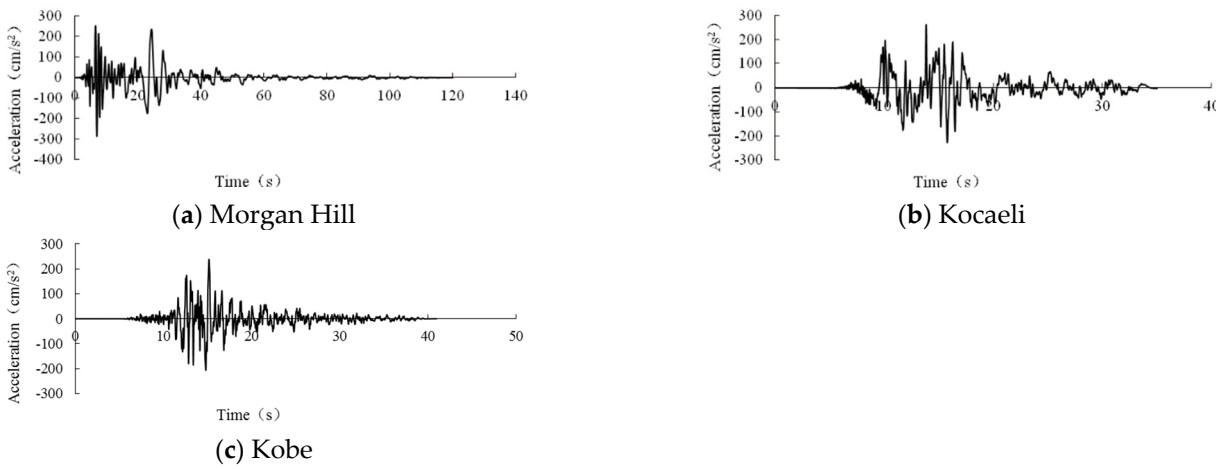

**Figure 3.** Earthquake time history.

### 4.3. PE Calculation and Analysis Results

The PE of model 1, model 2, model 3 and their corresponding first-mode equivalent SDOF systems were calculated using a MATLAB program. Figure 4 illustrates the comparison of the PE for model 1 and the first mode of its equivalent SDOF system, based on the three earthquake records with peak ground accelerations of 400 cm/s². The duration of each record was limited to the initial 40 s due to the strong vibrations. Figure 5 shows the comparison of the PE for model 2 and the first mode of its equivalent SDOF system based on the three records with a peak value of 400 cm/s².

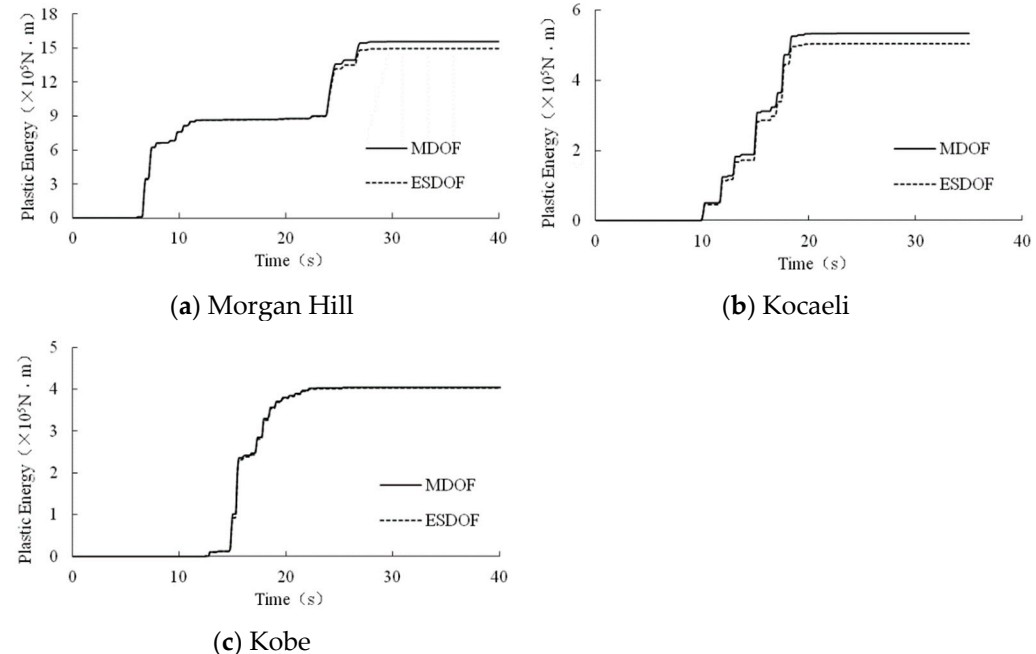

**Figure 4.** Comparison of PE between first-order-mode equivalent SDOF system and model 1.

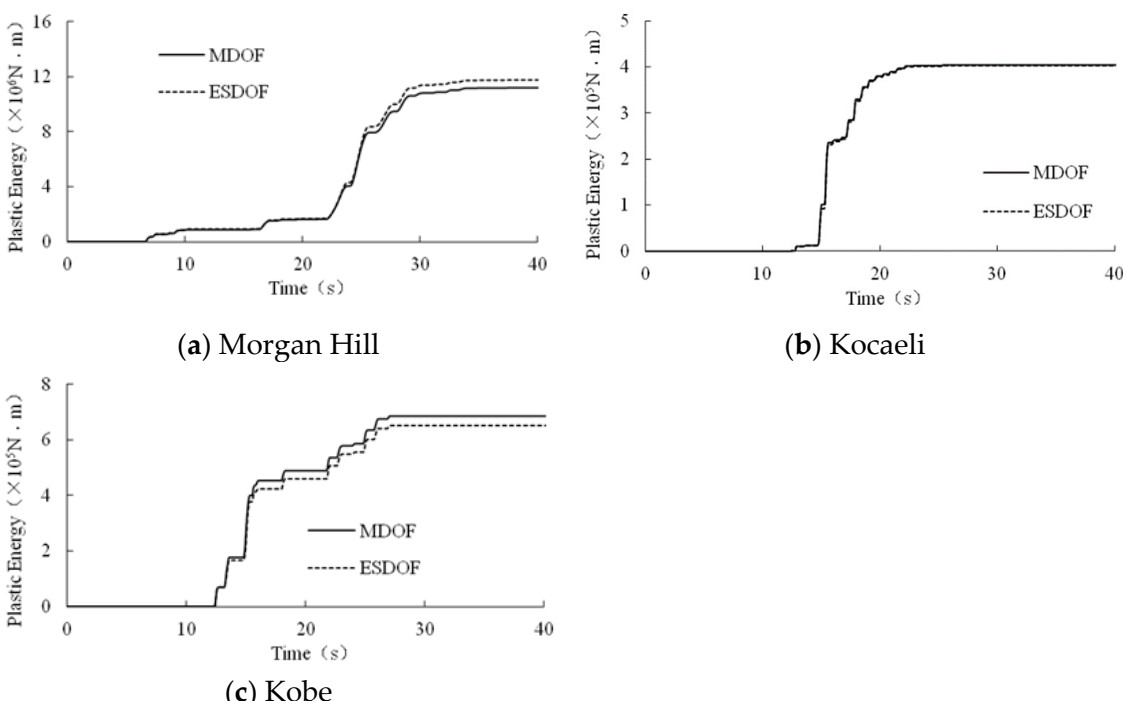

**Figure 5.** Comparison of PE between the structure and first-order-mode equivalent SDOF system.

As shown in Figures 4 and 5, model 1 and model 2, as well as their corresponding equivalent SDOF systems, exhibit similar trends across the various earthquake records considered, with similar PE values. At the initial stage of ground motions, the errors between the structures and their equivalent SDOF systems were small and increased over time until they stabilized. The results indicate that it is simple and accurate to estimate the plastic energy dissipation of a structure based on its first-order-mode equivalent SDOF system.

Figure 6 indicates the comparison of PE among the first four modes of its equivalent SDOF system for the El Centro earthquake. As shown in Figure 6, the value of PE for the fourth mode of the equivalent SDOF system was significantly smaller compared to those of the first three modes. Figure 7 compares the PE of the structure with the sum of the first three modes of the equivalent SDOF system. As shown in Figure 7, the structure and its equivalent SDOF system exhibit similar trends with similar PE values. The errors between them were small. The results indicate that it is simple and accurate to estimate the plastic energy dissipation of this high-rise structure by its first three order modes of the equivalent SDOF system.

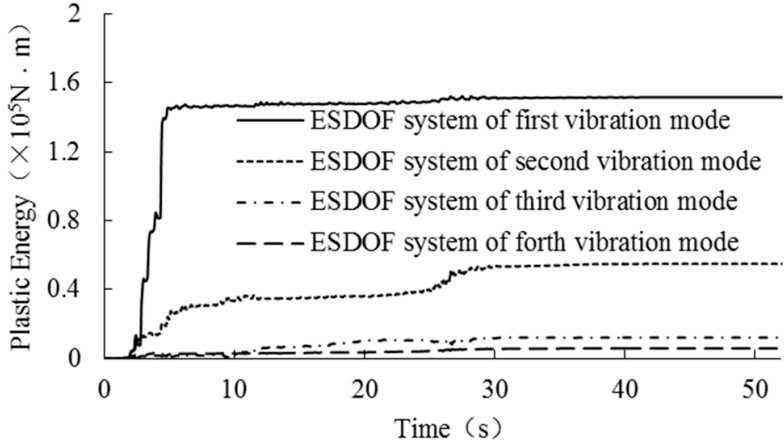

**Figure 6.** Time history curves of PE among ESDOF system of first four vibration modes (model 3).

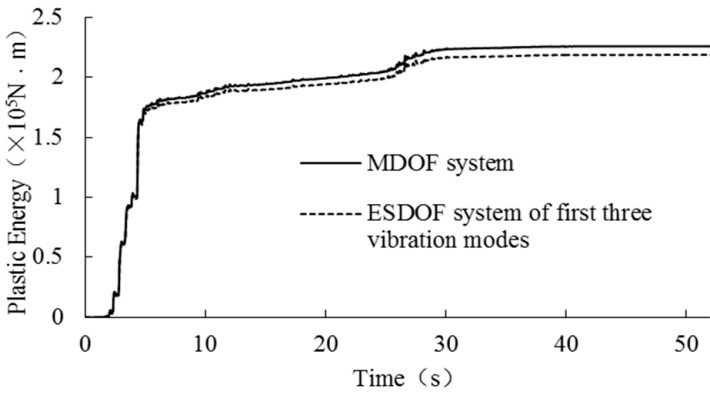

**Figure 7.** Comparison between the sum of PE of the first three ESDOF systems and the PE of model 3.

To investigate the impact of the peak value of the earthquake records on the proposed estimation method, the PE of model 1 and its equivalent SDOF system due to different peak values of the three ground accelerations are summarized in Table 3. As shown in Table 3, the Kocaeli earthquake resulted in the largest average error of PE between the two systems, with a value of 6.06%. The Kobe earthquake resulted in the smallest average error with 0.54%, and the Morgan Hill earthquake had an average error value of 4.2%. The ratio of the errors between the Kocaeli earthquake and the Kobe earthquake was 11.2.

**Table 3.** Comparison of peak values of PE between first-order-mode ESDOF system and model 1.

| Earthquake Records | Max. of PE and Error | Peak Values of Acceleration (cm/s$^2$) | | | | Average |
|---|---|---|---|---|---|---|
| | | **400** | **500** | **600** | **700** | **Values** |
| Morgan Hill | PE1 (kN·m) | 1497.3 | 2981.9 | 5241.7 | 8884.6 | |
| | PE$_S$ (kN·m) | 1557.0 | 3126.6 | 5478.6 | 9257.4 | |
| | PE1/PE$_S$ (%) | 96.17 | 95.37 | 95.68 | 95.97 | 95.80 |
| | Error (%) | 3.83 | 4.63 | 4.32 | 4.03 | 4.20 |
| Kocaeli | PE1 (kN·m) | 502.8 | 850.0 | 1239.3 | 1683.1 | |
| | PE$_S$ (kN·m) | 533.2 | 907.0 | 1323.2 | 1789.4 | |
| | PE1/PE$_S$ (%) | 94.31 | 93.72 | 93.66 | 94.06 | 93.94 |
| | Error (%) | 5.69 | 6.28 | 6.34 | 5.94 | 6.06 |
| Kobe | PE1 (kN·m) | 403.6 | 634.5 | 905.0 | 1208.8 | |
| | PE$_S$ (kN·m) | 405.6 | 639.4 | 908.0 | 1215.9 | |
| | PE1/PE$_S$ (%) | 99.517 | 99.235 | 99.678 | 99.416 | 99.46 |
| | Error (%) | 0.483 | 0.765 | 0.322 | 0.584 | 0.54 |

Note: PE$_S$ represents the plastic energy of the structure; PE1 is the plastic energy of the first-order-mode equivalent SDOF system.

The comparison of peak values of the PEs for model 2 and its first-order-mode equivalent SDOF system are presented in Table 4. Table 5 summarizes the PE of model 3 and different modes of the equivalent SDOF systems due to different peak values of the El Centro earthquake. As Table 4 shows, the Kocaeli earthquake resulted in the smallest average error of PE between the two systems and its value was 1.58%. The Kobe earthquake resulted in the greatest average error with 6.07%.

It can be seen from Tables 3–5 that the values of PE for both systems increased as the ground motion peak acceleration increased. However, the ratio between the sum of the first two modes of the equivalent SDOF system and the PE of the structure did not exhibit linearity with ground motion peak acceleration.

Table 4 indicates that the PE values of the equivalent SDOF system may be higher than those of the structure. This can be attributed to the assumption made for estimating PE using the equivalent SDOF system, where the displacement of the top floor of the structure is considered the generalized displacement. In reality, the displacement of a structure is induced by multiple modes, whereas only the displacement due to the first-order mode is considered and the other modes are ignored when calculating. Despite this limitation, the error was found to be acceptable.

**Table 4.** Comparison of peak values of PE between first-order-mode ESDOF system and model 2.

| Earthquake Records | Max. of PE and Error | Peak Values of Acceleration (cm/s$^2$) | | | | Average Values |
| --- | --- | --- | --- | --- | --- | --- |
| | | 400 | 500 | 600 | 700 | |
| Morgan Hill | PE1 (kN·m) | 11,766 | 16,954 | 22,415 | 28,101 | |
| | PE$_S$ (kN·m) | 11,225 | 16,030 | 21,076 | 26,310 | |
| | PE1/PE$_S$ (%) | 104.82 | 105.76 | 106.35 | 106.81 | 105.94 |
| | Error (%) | 4.82 | 5.76 | 6.35 | 6.81 | 5.94 |
| Kocaeli | PE1 (kN·m) | 3197.8 | 5766.6 | 8599.6 | 11,463 | |
| | PE$_S$ (kN·m) | 3303.2 | 5813.9 | 8511.0 | 11,250 | |
| | PE1/PE$_S$ (%) | 96.81 | 99.19 | 101.04 | 101.89 | 99.73 |
| | Error (%) | 3.20 | 0.82 | 1.04 | 1.89 | 1.58 |
| Kobe | PE1 (kN·m) | 651.9 | 1047.2 | 1481.8 | 1978.8 | |
| | PE$_S$ (kN·m) | 685.2 | 1114.0 | 1586.6 | 2123.1 | |
| | PE1/PE$_S$ (%) | 95.14 | 94.00 | 93.39 | 93.20 | 93.93 |
| | Error (%) | 4.86 | 6.00 | 6.61 | 6.80 | 6.07 |

**Table 5.** PE of ESDOF systems with different modes and model 3.

| Max. of PE and the Ratio | Peak Values of Acceleration (cm/s$^2$) | | | |
| --- | --- | --- | --- | --- |
| | 400 | 500 | 600 | 700 |
| PE2$_1$ (kN·m) | 151.8 | 285.5 | 455.1 | 669.0 |
| PE2$_2$ (kN·m) | 55.0 | 85.9 | 129.1 | 184.7 |
| PE2$_3$ (kN·m) | 12.2 | 16.8 | 22.8 | 30.2 |
| PE (kN·m) | 226.2 | 402.2 | 630.9 | 919.0 |
| (PE2$_1$ + PE2$_2$ + PE2$_3$)/PE (%) | 96.84 | 96.53 | 96.20 | 96.21 |

### 4.4. Comparison of the Two Estimation Methods

The two proposed plastic energy estimation methods were demonstrated using five different steel frame structures. These two estimation methods for ESDOF systems were established based on different principles. The method proposed in Section 3.1 was based on the assumption of ESDOF and considered only one mode shape, making it applicable only to a structure dominated by the first-order vibration mode. The method presented in Section 3.2 was based on the mode decomposition method and could be used for structures with many-order vibration modes.

The structure in Section 4.2 was used for the analysis. Four earthquake records were adopted based on site classification, including Chalfant Valley (1986.7.21, BISHOP-PARADISE LODGE, site I), Imperial Valley (1940.5.18, STATIONNO. 117, site II), San Fernando (1971.2.9, HOLLYWOOD STORAGE P.E. LOT, site III), and Kocaeli (1999.8.17, AMBARLI, site IV).

The values of peak ground acceleration were scaled to 400 cm/s$^2$, 500 cm/s$^2$, 600 cm/s$^2$, and 700 cm/s$^2$, and PE values were calculated for the structure using the two methods and compared with their real PE values. Table 6 shows the PE for the first mode of the ESDOF system estimated by two methods and the real PE of model 2 under different peak ground accelerations. It can be seen from Table 5 that the errors of the two methods were less than 8%. These results suggest that both methods can be used to estimate the PE of a structure that is dominated by one mode shape. The computational time of the method presented in Section 3.1 is almost half of that of the method presented in Section 3.2.

**Table 6.** Comparison of peak values of PE between the first-mode ESDOF system and model 2.

| Earthquake Records | Max. of PE and Ratio | Peak Values of Acceleration (cm/s$^2$) | | | | Computing Time (s) |
| --- | --- | --- | --- | --- | --- | --- |
| | | 400 | 500 | 600 | 700 | |
| Chalfant Valley | PE (kN·m) | 151.37 | 286.39 | 441.77 | 601.37 | |
| | PE1 (kN·m) | 148.81 | 285.02 | 447.17 | 611.38 | 9.813 |
| | PE2$_1$ (kN·m) | 150.03 | 285.06 | 443.89 | 607.36 | 16.261 |
| | Error of PE1 and PE (%) | 1.69 | 0.48 | 1.22 | 1.67 | |
| | Error of PE2$_1$ and PE (%) | 0.89 | 0.47 | 0.48 | 1.01 | |

**Table 6.** *Cont.*

| Earthquake Records | Max. of PE and Ratio | Peak Values of Acceleration (cm/s$^2$) | | | | Computing Time (s) |
|---|---|---|---|---|---|---|
| | | 400 | 500 | 600 | 700 | |
| Imperial Valley | PE (kN·m) | 1595.3 | 2532.4 | 3636.9 | 4818.9 | 1.820 4.164 |
| | PE1 (kN·m) | 1552.3 | 2450.4 | 3519.4 | 4677.3 | |
| | PE2$_1$ (kN·m) | 1583.3 | 2506.7 | 3609.9 | 4817.5 | |
| | Error of PE1 and PE (%) | 2.69 | 3.23 | 3.23 | 2.94 | |
| | Error of PE2$_1$ and PE (%) | 0.75 | 1.01 | 0.74 | 0.03 | |
| San Fernando | PE (kN·m) | 224.60 | 608.90 | 1152.1 | 1810.2 | 2.602 5.369 |
| | PE1 (kN·m) | 219.67 | 600.16 | 1164.2 | 1853.7 | |
| | PE2$_1$ (kN·m) | 214.13 | 594.47 | 1119.9 | 1750.3 | |
| | Error of PE1 and PE (%) | 2.19 | 1.43 | 1.05 | 2.40 | |
| | Error of PE2$_1$ and PE (%) | 4.66 | 2.37 | 2.79 | 3.31 | |
| Kocaeli | PE (kN·m) | 485.41 | 907.72 | 1453.6 | 2135.8 | 121.719 234.790 |
| | PE1 (kN·m) | 468.86 | 882.66 | 1419.1 | 2100.0 | |
| | PE2$_1$ (kN·m) | 469.43 | 867.37 | 1367.3 | 1984.7 | |
| | Error of PE1 and PE (%) | 3.41 | 2.76 | 2.37 | 1.67 | |
| | Error of PE2$_1$ and PE (%) | 3.29 | 4.45 | 5.94 | 7.07 | |

## 5. Conclusions

In this paper, two methods for estimating a structure's PE based on FAM have been proposed. Method 1 is based on the assumption of the ESDOF system. Method 2 is based on the mode decomposition method and suggests estimating the PE of an MDOF system by summing the PEs of several-order modes for an equivalent SDOF system. Based on the numerical simulations conducted in this study, the following conclusions can be drawn:

1. Method 1 can be used only for structures where the seismic response is dominated by the ith mode shape. The PE of the MDOF system can be estimated by multiplying the PE for the corresponding order mode of the ESDOF system by the participation coefficient of that order mode shape. The generalized displacement of this method can use the displacement of the top floor of the structure, making it a simple and efficient method.
2. Method 2 can be used for estimating the PE of multi-story or high-rise structures by using the sum of the PE for the first two or three modes of the ESDOF system. The errors between the two systems are small.
3. The values of PE for both the MDOF and ESDOF systems increase with the enlargement in ground motion peak acceleration. However, the ratio between the sum of PE for the first several modes of the ESDOF system and the PE of the structure does not show linearity with the ground motion peak acceleration. However, the errors are all acceptable in actual projects.

**Author Contributions:** Conceptualization, J.Q.; Methodology, Y.M.; Software, Y.S.; Validation, Y.T. All authors have read and agreed to the published version of the manuscript.

**Funding:** This research was funded by Ministry of Education key laboratory of Facility Fishery-Dalian Ocean University (Ministry of Education of the People's Republic of China); National Natural Science Foundation of China (grant number-51778113); Scientific Research Project of the Educational Department of Liaoning Provincial (The Education Department of Liaoning Province, grant number-100920202019).

**Data Availability Statement:** Not applicable.

**Conflicts of Interest:** The authors declare no conflict of interest.

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
