# Peer review of "Simplified Estimation Method of Plastic Energy Dissipation for MDOF Systems Using Force Analogy Method"

_buildings, doi:10.3390/buildings13051330_

Round 1

Reviewer 1 Report

The paper presents two methods for estimating plastic energy dissipation in inelastic structures under seismic action - equivalent SDOF and MDOF - and carries out numerical simulations. The methods have practical engineering applications for the seismic design of structures and are in line with the scope of this journal. The authors have adequately explained the purpose of their research, which means that the articles are well organised and written. The reviewers appreciated the innovative approach presented in the manuscript. However, the reviewers had the following comments and questions:

1. Only two references from the last five years are cited in the introductory section of the article (Citations 17 and 18). It is suggested that the authors could add some relevant studies from recent years, providing an overview of the need for current research, the shortcomings of existing methods and the current level of development of the problem.

2. Line 107, is the derivation of the total bending moment. However, Substituting (8) and (9) into (3) does not seem to lead to (10), should it be amended to "Substitute (8) and (9) into (5) to obtain the total bending moment equation"?

3. The expression of the total bending moment in formula (10) is inconsistent with that in formula (12).

4. In line 241-250, the authors interpret Figure 6 as ‘As shown in Fig.6, the value of PE for the 4th mode of the equivalent SDOF system was significantly smaller compared to those of the first three modes .’ whereas in Fig.6, the value of PE for the both 3th and 4th are significantly smaller compared to the 1th and 2th, and how the conclusion that ‘take the first three order modes as a result', further argumentation is needed.

5. The author mentions "average error" in the explanation of Tables 3 and 4, but this value is not reflected in the table. For better understanding by the reader, it is suggested that a column could be added to the tables to reflect this value.

The reviewers consider that the acceptance of the manuscript depends on the revisions. Authors need to respond point by point and provide a rebuttal if some of the reviewers' comments cannot be revised.

  • The author's English writing has correct grammar and is fluent in expression.

  •  
  •  

Author Response

Reply to Comments of Reviewers

We thank the editor for offering us the comments. We believe that these comments have enabled us to improve our manuscript. In the following, we offer item-to-item reply to each comment and list the associated changes in the manuscript to address the comment. For clarity, the changes of revision are marked up using the “Track Changes” function in revised version.

Replies to reviewer’s comments:

Reviewer #1:

  1. Only two references from the last five years are cited in the introductory section of the article (Citations 17 and 18). It is suggested that the authors could add some relevant studies from recent years, providing an overview of the need for current research, the shortcomings of existing methods and the current level of development of the problem.

Reply:

We gratefully appreciate for your valuable comment. Four references are included in the introductory section. The development of structural analysis with material nonlinear behavior based on FAM is accurate and efficient available in the literature mentioned in the third paragraph of Section 1.

  1. 2. Line 107, is the derivation of the total bending moment. However, Substituting (8) and (9) into (3) does not seem to lead to (10), should it be amended to "Substitute (8) and (9) into (5) to obtain the total bending moment equation"?

Reply:

We appreciate the reviewer’s important suggestion and apologize for the omission of the derivation and expression of the formulae. In the revised manuscript, the expression of the formula (10) has been modified.

Line 111 of Section2

Substituting equations (8) and (9) into equation (5), the total moment vector is

      (10)

  1. The expression of the total bending moment in formula (10) is inconsistent with that in formula (12).

Reply:

We appreciate your valuable suggestion and apologize for the omission of the expression of the formulae. In the revised manuscript, the expression of the formula (10) has been modified.

  1. In line 241-250, the authors interpret Figure 6 as ‘As shown in Fig.6, the value of PE for the 4th mode of the equivalent SDOF system was significantly smaller compared to those of the first three modes.’ whereas in Fig.6, the value of PE for the both 3th and 4th are significantly smaller compared to the 1th and 2th, and how the conclusion that ‘take the first three order modes as a result', further argumentation is needed.

Reply:

Thanks for your suggestion. If only the 1st and 2nd modes are considered, we could obtain the corresponding plastic energies of the structure and the plastic energy dissipation of the 1st order mode shape of an equivalent SDOF system as the following Table 0 shown. The error is about 7~8.5%. When the first three order modes are taken as a result, the error is less than 4% as shown in Table 5 in the paper. As far as the error is concerned, the first three order modes are suggested to be considered.

Table 0. PE of ESDOF systems with different modes and model 3.

Max. of PE and the ratio

Peak values of acceleration(cm/s2

400

500

600

700

PE21 (kN﹒m)

151.8

285.5

455.1

669.0

PE22 (kN﹒m)

55.0

85.9

129.1

184.7

PE (kN﹒m)

226.2

402.2

630.9

919.0

(PE21+ PE22)/ PE (%)

91.42

92.34

92.60

92.89

  1. 5. The author mentions "average error" in the explanation of Tables 3 and 4, but this value is not reflected in the table. For better understanding by the reader, it is suggested that a column could be added to the tables to reflect this value.

Reply:

We truly appreciate the reviewer’s valuable comment. The Tables 3 and 4 are revised in the manuscript.

It should be better to show the ratios because this should make immediately clear that some of the errors are negative and some are positive, especially in Table 4. So, the ratios are given in Tables 3 and 4 as same as Table 5.

Table 3. Comparison of peak values of PE between 1st order mode ESDOF system and model 1.

Earthquake    

Records

Max. of PE, ratio, and error

Peak values of acceleration(cm/s2

Average

400

500

600

700

values

Morgan Hill

PE1(kN﹒m)

1497.3

2981.9

5241.7

8884.6

PES(kN﹒m)

1557.0

3126.6

5478.6

9257.4

PE1 / PES (%)

96.17

95.37

95.68

95.97

95.80

Error (%)

3.83

4.63

4.32

4.03

4.20

Kocaeli

PE1(kN﹒m)

502.8

850.0

1239.3

1683.1

PES(kN﹒m)

533.2

907.0

1323.2

1789.4

PE1 / PES (%)

94.31

93.72

93.66

94.06

93.94

Error (%)

5.69

6.28

6.34

5.94

6.06

Kobe

PE1(kN﹒m)

403.6

634.5

905.0

1208.8

PES(kN﹒m)

405.6

639.4

908.0

1215.9

PE1 / PES(%)

99.517

99.235

99.678

99.416

99.46

Error (%)

0.483

0.765

0.322

0.584

0.54

Table 4. Comparison of peak values of PE between 1st order mode ESDOF system and model 2.

Earthquake    

Records

Max. of PE, ratio, and error

Peak values of acceleration(cm/s2

Average

values

400

500

600

700

Morgan Hill

PE1(kN﹒m)

11766

16954

22415

28101

PES(kN﹒m)

11225

16030

21076

26310

PE1 / PES (%)

104.82

105.76

106.35

106.81

105.94

Error (%)

4.82

5.76

6.35

6.81

5.94

Kocaeli

PE1(kN﹒m)

3197.8

5766.6

8599.6

11463

PES(kN﹒m)

3303.2

5813.9

8511.0

11250

PE1 / PES (%)

96.81

99.19

101.04

101.89

99.73

Error (%)

3.20

0.82

1.04

1.89

1.58

Kobe

PE1(kN﹒m)

651.9

1047.2

1481.8

1978.8

PES(kN﹒m)

685.2

1114.0

1586.6

2123.1

PE1 / PES (%)

95.14

94.00

93.39

93.20

93.93

Error (%)

4.86

6.00

6.61

6.80

6.07

Reviewer 2 Report

The research is adequately designed and the article is well written and organized; however, a few comments to the article can be found in the attached file.

Author Response

Reply to Comments of Reviewers

We thank the editor for offering us the comments. We believe that these comments have enabled us to improve our manuscript. In the following, we offer item-to-item reply to each comment and list the associated changes in the manuscript to address the comment. For clarity, the changes of revision are marked up using the “Track Changes” function in revised version.

Replies to reviewer’s comments:

Reviewer #2:

  1. When first used, the acronym FAM should be defined.

Reply:

Thanks for your suggestion. We apologize for the omission. In the revised manuscript, the force analogy method is added in the line 55.

  1. 2. The authors write «Substituting equations (8) and (9) into equation (3)», but it seems that instead of «equation (3)» they should write “equation (5)”.

Reply:

We appreciate the reviewer’s important suggestion and apologize for the omission of the derivation and expression of the formula. In the revised manuscript, the expression of the formula (10) has been modified.

Line 111 of Section2

Substituting equations (8) and (9) into equation (5), the total moment vector is

      (10)

  1. The authors write «premultiplying it by Φi». They should specify the are multiplying on the left by Φit.

Reply:

We appreciate your valuable suggestion. In the revised manuscript, the expression has been modified in the line 141.

  1. The formula (31) seems to be incorrect: Q’’ is repeated twice and Q’ is missing.

Reply:

We truly appreciate the reviewer’s valuable comment. In the revised manuscript, the formula (31) has been modified.

  1. 5. In the Line 164, the authors should specify they are multiplying on the left.

Reply:

We appreciate your valuable suggestion. In the revised manuscript, the expression has been modified in the line 167.

  1. 6. The authors write «ith», but they probably meant «1st».

Reply:

Thanks for your suggestion. We apologize for the careless writing. In the revised manuscript, the mistake has been modified in the line 244.

  1. The authors write that «Figure 6 indicates the comparison of PE between model 3 and the first four modes of its equivalent SDOF system», however it seems that there only the PE of the first 4 modes are represented, and not that of the MDOF of model 3.

Reply:

We truly appreciate the reviewer’s valuable comment. In the revised manuscript, the expression has been modified in the line 246.

Line 246 of Section 4.3

Figure 6 indicates the comparison of PE among the first four modes of its equivalent SDOF system for the El Centro earthquake.

  1. It’s not clear why in tables 3 and 4 the authors report the % error, while in table 5 the authors report the ratios. In particular, it should be better to show the ratios in table 4, because this should make immediately clear that some of the errors are negative and some are positive.

Reply:

We appreciate your valuable suggestion. The Tables 3 and 4 are revised in the manuscript.

Table 3. Comparison of peak values of PE between 1st order mode ESDOF system and model 1.

Earthquake    

Records

Max. of PE, ratio, and error

Peak values of acceleration(cm/s2

Average

400

500

600

700

values

Morgan Hill

PE1(kN﹒m)

1497.3

2981.9

5241.7

8884.6

PES(kN﹒m)

1557.0

3126.6

5478.6

9257.4

PE1 / PES (%)

96.17

95.37

95.68

95.97

95.80

Error (%)

3.83

4.63

4.32

4.03

4.20

Kocaeli

PE1(kN﹒m)

502.8

850.0

1239.3

1683.1

PES(kN﹒m)

533.2

907.0

1323.2

1789.4

PE1 / PES (%)

94.31

93.72

93.66

94.06

93.94

Error (%)

5.69

6.28

6.34

5.94

6.06

Kobe

PE1(kN﹒m)

403.6

634.5

905.0

1208.8

PES(kN﹒m)

405.6

639.4

908.0

1215.9

PE1 / PES(%)

99.517

99.235

99.678

99.416

99.46

Error (%)

0.483

0.765

0.322

0.584

0.54

Table 4. Comparison of peak values of PE between 1st order mode ESDOF system and model 2.

Earthquake    

Records

Max. of PE, ratio, and error

Peak values of acceleration(cm/s2

Average

values

400

500

600

700

Morgan Hill

PE1(kN﹒m)

11766

16954

22415

28101

PES(kN﹒m)

11225

16030

21076

26310

PE1 / PES (%)

104.82

105.76

106.35

106.81

105.94

Error (%)

4.82

5.76

6.35

6.81

5.94

Kocaeli

PE1(kN﹒m)

3197.8

5766.6

8599.6

11463

PES(kN﹒m)

3303.2

5813.9

8511.0

11250

PE1 / PES (%)

96.81

99.19

101.04

101.89

99.73

Error (%)

3.20

0.82

1.04

1.89

1.58

Kobe

PE1(kN﹒m)

651.9

1047.2

1481.8

1978.8

PES(kN﹒m)

685.2

1114.0

1586.6

2123.1

PE1 / PES (%)

95.14

94.00

93.39

93.20

93.93

Error (%)

4.86

6.00

6.61

6.80

6.07

  1. The authors write about «the subscripts d and r», but in the previous formula (3) the subscript r doesn’t appear. Probably all this paragraph should be shifted after formula (4).

Reply:

We truly appreciate the reviewer’s valuable comment. In the revised manuscript, the paragraph has been modified and the introduction about the subscripts d and r is shifted after formula (4).

Line 96~101

Where Il is a column vector, and its elements are all 1; ag is the acceleration of the ground. represents the elastic stiffness matrix of a structure after performing static condensation,

,                              (4)

Where the subscripts d and r indicate the matrices or vectors corresponding to the displacement DOFs and the rotation DOFs, respectively; for an n DOF system, n=d+r.

  1. 10. The word «For» is capitalized even if it’s preceded by a semicolon.

Reply:

We appreciate your valuable suggestion. The first letter of the word “For” is converted to lowercase in the revised manuscript in the line 101.

  1. 11. The authors write «restoring forces and FR(t)». It seems that the word “and” shouldn’t be

there.

Reply:

We appreciate your valuable suggestion. In the revised manuscript, the expression has been modified in the line 101.

  1. The authors write «see Qu, 2013», while they should use a different reference system ([#of the reference]). In addition, it seems that the cited reference is missing in the references paragraph.

Reply:

We appreciate the reviewer’s important suggestion and apologize for the mistake. In the revised manuscript, the expression has been modified.

Line 115~116

The energy equation can be derived based on the FAM (discussed in details in the reference [23]) as

  1. 13. Figure 2 is shifted between the “title” «Model 3» and the paragraph in which the description of the model is.

Reply:

We appreciate the reviewer’s important suggestion and apologize for the sequence confusion of the paragraph and figure. In the revised manuscript, the position of Figure 2 has been modified in the line 213.

14.In the caption of figure 5 some spaces are missing: «structureand1st».

Reply:

Thanks for your suggestion. We apologize for the careless writing. In the revised manuscript, the misspelling has been modified in the line 238.

15.The caption of fig. 6 should specify that it’s referred to model 3.

Reply:

We appreciate the reviewer’s important suggestion and apologize for the omission. The caption of Fig. 6 has been modified in the line 255 in the revised manuscript.

Line 255

Figure 6. Time history curves of PE among ESDOF system of first four vibration modes (model 3).

Reviewer 3 Report

The purpose of this study is to establish formulas of plastic energy of MDOF systems and their equivalent SDOF systems using the FAM based on the assumption of the equivalent SDOF system and the mode decomposition method. The simulation results demonstrate that the proposed methods and formulas accurately estimate plastic energy dissipation in multi-story and high-rise structures, while also requiring fewer calculations and less storage. The research on the subject of this paper is of practical value in construction engineering, but there are still deficiencies in the paper that need to be modified:

1.Most of the references in this paper are too old and many of them are ten years old, it is suggested that the authors should refer more to the new progress in related fields in recent years to highlight the significance of the research.

2.At the end of the introduction, the emphasis should highlight the significance and value of establishing the plastic performance formula of the MDOF system and its equivalent SDOF system, which is not clearly reflected in the article.

3.In line 14, the noun phrase single degree seems to be missing a determiner before it. Consider adding an article

4.In line 34, the word elasto-plasticshould be changed to elastic-plastic or elastoplastic.Please check and modify the other locations

5.In line 217, the table names and tables should be optimally laid out and properly centered, and the authors are requested to check whether the rest of the table names are reasonably placed.

6.In line 209, figure names should be properly centered, please check if the rest of the figure names are reasonably positioned.

7."References" should be centered.

8.The serial number of the reference should be indicated by a number in square brackets, e.g.[1].

Author Response

Reply to Comments of Reviewers

We thank the editor for offering us the comments. We believe that these comments have enabled us to improve our manuscript. In the following, we offer item-to-item reply to each comment and list the associated changes in the manuscript to address the comment. For clarity, the changes of revision are marked up using the “Track Changes” function in revised version.

Replies to reviewer’s comments:

Reviewer #3:

  1. Most of the references in this paper are too old and many of them are ten years old, it is suggested that the authors should refer more to the new progress in related fields in recent years to highlight the significance of the research.

Reply:

We gratefully appreciate for your valuable comment. Four references from the last five years are included in the introductory section. The development of structural analysis with material nonlinear behavior based on FAM is accurate and efficient available in the literature mentioned in the third paragraph of Section 1.

3rd paragraph of Section 1

Over time, many scholars have not only worked to improve the FAM theory [24], but also explored the application of this method [25-27]. Hao et al. [28] presented the static pushover analysis for nonlinear fiber beam element conducted on the foundation of the FAM and the results shows the algorithm complexity of the proposed method is decreased about 80%, and its computing efficiency is increased at least five times. The accumulation of research in this area indicates that the FAM is a reliable, accurate, and efficient method for analysis, since it only requires the initial stiffness of structures. 

References:

19. Mohammed Samier Sebaq, Ying Zhou, Ge Song, Yi Xiao. Plastic energy evaluation of bilinear SDOF systems with fluid viscous dampers[J]. Struct Design Tall Spec Build, 2023, 10.1002/tal.2011.

26. Li Hongnan, Song Jianzhu, Li Gang. Study on failure mode of nonlinear vibration device based on FAM-MMBC for MDOF system[J]. Journal of Building Structures, 2017,38(1): 93-98.

27. Li Gang, Yu Dinghao. Efficient inelasticity-separated finite element method for material nonlinearity analysis[J]. ASCE, Journal of Engineering Mechanics, 2018,144(4):0401800.

28. Hao Runxia, Wang Mouting, Jia Shuo, Li Gang. Static pushover analysis of frame structures based on force analogy method[J]. Journal of Southwest Jiao Tong University, 2020,55(5): 1028-1035.

  1. 2. At the end of the introduction, the emphasis should highlight the significance and value of establishing the plastic performance formula of the MDOF system and its equivalent SDOF system, which is not clearly reflected in the article.

Reply:

We appreciate the reviewer’s important suggestion. In order to highlight the significance and value of this study, the following sentences are adding at the end of the introduction in the revised manuscript.

Last paragraph in Section 1

This research aims to propose two estimation methods for the plastic energy dissipation of a MDOF system, which can serve as a useful tool to analyze structural damage for energy-based seismic design. In seismic design, the inelastic energy spectrum is often used, and by using the proposed method plastic energy spectrum can be obtained and suitable for multi-degree of freedom systems, which will be introduced in another research article in the future.

  1. In line 14, the noun phrase single degree seems to be missing a determiner before it. Consider adding an article.

Reply:

We appreciate your valuable suggestion. In the revised manuscript, the expression has been modified in line 15.

Line 15

While a single degree of freedom (SDOF) system provides a simple and effective method for estimating plastic energy dissipation, ….

  1. In line 34, the word “elasto-plastic” should be changed to “elastic-plastic” or “elastoplastic”. Please check and modify the other locations.

Reply:

Thanks for your suggestion. We apologize for the careless writing. In the revised manuscript, the mistake has been modified in the line 35.

  1. 5. In line 217, the table names and tables should be optimally laid out and properly centered, and the authors are requested to check whether the rest of the table names are reasonably placed.

Reply:

We appreciate your valuable suggestion. In the revised manuscript, we checked and corrected table formatting issues according to the template in the text.

  1. 6. In line 209, figure names should be properly centered, please check if the rest of the figure names are reasonably positioned.

Reply:

Thanks for your suggestion. We apologize for the careless format issues. In the revised manuscript, we checked and corrected figure formatting issues according to the template in the text.

  1. “References” should be centered.

Reply:

We appreciate the reviewer’s important suggestion. At present, “References” is placed on the left according to the template. If further modifications are needed, please let us know. Thank you very much.

  1. The serial number of the reference should indicated by a number in square brackets, e.g.[1].

Reply:

We gratefully appreciate for your valuable comment. In the text, reference numbers have been placed in square brackets,numbers of “References” are lists at end of text. If further modifications are needed, it is our honor with your important suggestion.

Round 2

Reviewer 1 Report

I agree with the publication of the article.

The author's English writing has correct grammar and is fluent in expression.